# Signatures of a magnetic-field-induced Lifshitz transition in the ultra-quantum limit of the topological semimetal ZrTe$_5$

S. Galeski [1,2] ✉, H. F. Legg [3] ✉, R. Wawrzyńczak [1], T. Förster[4], S. Zherlitsyn[4], D. Gorbunov[4], M. Uhlarz[4], P. M. Lozano [5], Q. Li[5], G. D. Gu[5], C. Felser [1], J. Wosnitza[4,6], T. Meng [7] & J. Gooth [1,2] ✉

The quantum limit (QL) of an electron liquid, realised at strong magnetic fields, has long been proposed to host a wealth of strongly correlated states of matter. Electronic states in the QL are, for example, quasi-one dimensional (1D), which implies perfectly nested Fermi surfaces prone to instabilities. Whereas the QL typically requires unreachably strong magnetic fields, the topological semimetal ZrTe$_5$ has been shown to reach the QL at fields of only a few Tesla. Here, we characterize the QL of ZrTe$_5$ at fields up to 64 T by a combination of electrical-transport and ultrasound measurements. We find that the Zeeman effect in ZrTe$_5$ enables an efficient tuning of the 1D Landau band structure with magnetic field. This results in a Lifshitz transition to a 1D Weyl regime in which perfect charge neutrality can be achieved. Since no instability-driven phase transitions destabilise the 1D electron liquid for the investigated field strengths and temperatures, our analysis establishes ZrTe$_5$ as a thoroughly understood platform for potentially inducing more exotic interaction-driven phases at lower temperatures.

Magnetic fields are an important experimental tuning knob for dimensional reduction. This happens at sufficiently strong fields when all electrons are confined to the lowest Landau level−a regime known as the ultra-quantum limit (UQL)[1]. Given that the Landau bands in three-dimensional (3D) systems only disperse with the momentum parallel to the field, electrons within the UQL form a liquid occupying only a single quasi-1D band. Such liquids are known to be on the verge of instabilities and even small perturbations can cause them to undergo a phase transition[2]. It was consequently predicted that a 3D electron gas exposed to high enough magnetic fields can exhibit a plethora of exotic ground states such as Wigner crystals or exotic varieties of charge and spin density waves that could exhibit a quantised Hall effect[3–6].

In most materials, the quantum limit cannot be reached since it requires too large magnetic fields. Certain low-density semimetals, however, have been shown to reach the UQL at experimentally accessible fields. With some of the the most prominent examples being graphite[7], bismuth[8], NbP[9], and TaAs[10] (quantum limits in the range 10–30 T), but a thorough understanding of their field-induced phases remains a difficult task due to the intrinsic complexities of those materials. Searching for a simpler material, ideally tuneable in situ, in which the UQL can be reached by available magnetic fields is, therefore, an important goal.

[1]Max Planck Institute for Chemical Physics of Solids, Nöthnitzer Straße 40, 01187 Dresden, Germany. [2]Physikalisches Institut, Universität Bonn, Nussallee 12, 53115 Bonn, Germany. [3]Department of Physics, University of Basel, Klingelbergstrasse 82, CH-4056 Basel, Switzerland. [4]Hochfeld-Magnetlabor Dresden (HLD-EMFL) and Würzburg-Dresden Cluster of Excellence ct.qmat, Helmholtz-Zentrum Dresden-Rossendorf, 01328 Dresden, Germany. [5]Condensed Matter Physics and Materials Science Department, Brookhaven National Laboratory, Upton, NY, USA. [6]Institut für Festkörper- und Materialphysik, Technische Universität Dresden, 01069 Dresden, Germany. [7]Institute of Theoretical Physics and Würzburg-Dresden Cluster of Excellence ct.qmat, Technische Universität Dresden, 01069 Dresden, Germany. ✉e-mail: stanislaw.galeski@cpfs.mpg.de; henry.legg@unibas.ch; johannes.gooth@cpfs.mpg.de

The recent surge in both experimental and theoretical studies of Dirac and Weyl semimetals has also sparked renewed interest in field-induced effects in those systems. One prominent example is the pentatelluride material family of $ZrTe_5$ and $HfTe_5$. These compounds are close to the transition between 3D strong and weak topological insulator (TI)[11]. $ZrTe_5$ has, therefore, recently risen to prominence as an excellent solid-state realisation of the (weakly gapped) 3D Dirac Hamiltonian[12–17]. Because the chemical potential typically lies outside the gap, $ZrTe_5$ in practice is a diamagnetic metal[18]. It shows no signatures of magnetic interactions and is in that sense a fairly "clean" electronic material.

In agreement with a Dirac model, at low temperatures our samples exhibit a single electron-type Fermi surface at the gamma point comprising of less than 1% of the Brillouin zone, and a charge-carrier concentration in the range $10^{16}$–$10^{17}$ $cm^{-3}$.[15] Such small charge-carrier densities enable entry into the quantum limit at magnetic fields of the order of 1 T, enabling, to the best of our knowledge, the study of a system deeper in the quantum limit than in previously studied 3D materials. In addition, recent studies of $ZrTe_5$ and $HfTe_5$ have revealed that, due to the exceptional purity of the available samples (electron mobilities of the order of 400,000 $cm^2/Vs$[15]) and simplicity of the band structure, most physical low-field ($B < 2$ T) properties can be described with simple linear-response calculations down to 2 K. As such, both $ZrTe_5$ and $HfTe_5$ are excellent model systems for the study of Dirac electrons deep in the ultra-quantum limit[15].

Previous studies of interactions in $ZrTe_5$ in magnetic fields, done by various groups, found indications of both interaction-driven gaps and samples that appear gapless. Tang et al., for example, identified a metal-insulator transition at about 4 T in samples with a small charge-carrier density with a quantum limit at 1.2 T[19]. This transition is associated with a quantisation of the Hall conductivity, which the authors identify as a possible realisation of the 3D quantum Hall effect[4]. Data with similar appearance, although with higher charge-carrier density and thus enhanced magnetic-field scales, have been interpreted by Liu et al. as evidence for an interaction-driven mass generation (density wave formation)[20]. In contrast, our own previous experiments have shown evidence for a gapless electronic system without density waves, but still exhibiting a quasi-quantised Hall conductivity[15]. One possible explanation for this discrepancy is that $ZrTe_5$ might actually be close to interaction-driven instabilities, with sample and experiment-specific variations pushing the system to one state or the other.

This motivated us to further characterise the properties of $ZrTe_5$ in high magnetic fields. To that end, we have performed magnetoresistance and ultrasound-propagation experiments in pulsed magnetic fields up to 64 T on two samples entering the quantum limit at 1.2 and 0.6 T, respectively, with the field applied along the crystallographic b-axis. As is well known from highly oriented graphite (HOPG), the combination of transport with ultrasound measurements is particularly helpful for identifying phase transitions. In HOPG, measurements of sound propagation and attenuation for example revealed a cascade of well-pronounced peaks signalling a series of field-induced electronic phase transitions, while the magnetoresistance of the same samples merely showed a broad anomaly with small kinks at the phase transitions - on the edge of the experimental resolution[7].

We find that the presently analysed samples do not undergo any thermodynamic phase transition. Instead, the Zeeman effect in conjunction with the massive Dirac nature of electrons in $ZrTe_5$ lead to a field-induced Lifshitz transition. The latter results in the formation of two 1D Weyl points. In addition, the Hall coefficient suggests that the system can be tuned to exhibit perfect charge neutrality at 25 and 8 T, respectively, in these samples.

## Results and discussion

Our study reports on $ZrTe_5$ samples similar to those studied in refs. 15,16 (charge-carrier density $n = 6.1 \times 10^{16}$ $cm^{-3}$), grown by the tellurium-flux method. In order to investigate possible magnetic field-induced phases we first performed temperature-dependent magnetoresistance and Hall measurements in pulsed fields up to 64 T at the Dresden High Magnetic Field Laboratory (HZDR). These measurements are shown in Fig. 1a. An important feature of our data is a rapid increase of the longitudinal resistance once the sample enters the quantum limit, which then almost saturates for fields above ca. 35 T. These observations agree with previous measurements and would also be consistent with the onset of a metal-insulator-transition. Indeed, the resistance at 60 T increases by almost 50.000% as compared to the zero-field value. Also, the Hall data seems to be consistent with the opening of a spectral gap due to the formation of a charge density wave (CDW). Our measurements of the high-field Hall effect, shown in Fig. 1a, exhibit a striking change of the Hall coefficient above 10 T, with the Hall resistivity beginning to decrease with increasing field to the point where it changes sign at about 25 T. Here we have taken the convention that the positive Hall resistance is electron-like for positive fields. Similar behaviour has been previously reported in a range of CDW materials with field-induced transitions[21,22] and in the Weyl semimetal TaP where the change of sign of the Hall effect has been to attributed to annihilation of Weyl nodes and opening a spectral gap[23].

However, as outlined above, transport alone can be insufficient to prove or disprove the existence of a density wave. Therefore, we have performed additional ultrasound-propagation experiments in pulsed magnetic fields up to 56 T. Such measurements provide an independent, and usually very sensitive, test for bulk phase transitions: the velocity of sound propagating through a crystal is proportional to the elastic modulus and thus a thermodynamic quantity[24]. Changes of the sound velocity reflect changes to the overall free energy of the coupled electron-phonon system. Sudden changes in the free energy, such as those caused by a gap opening related to a phase transition, should, therefore, have a significant effect on the sound velocity, especially when accompanied by breaking of translational invariance such as in the case of charge and spin density waves[25,26].

Figure 1b shows the field dependence of the sound velocity of a transverse phonon mode polarised along the b-axis and of a longitudinal phonon, both propagating along the crystallographic a-axis with magnetic field parallel to the b-axis. The field dependence of the speed of sound exhibits a softening of both elastic modes at fields above 4 T and saturates above about 15 T, with a shallow minimum observable at about 10 T in the case of the longitudinal mode. Unlike what would be expected for a phase transition, however, both curves appear smooth without any sharp features that could be interpreted as the onset of such a transition.

This finding prompts us to reconsider our transport data measured at high fields. To avoid the complication that $\rho_{xx}$ is a mixture of longitudinal and transverse conductivities, we have inverted the resistance tensor and extracted the longitudinal conductivity $\sigma_{xx}(B)$ and Hall conductivity $\sigma_{xy}(B)$. Figure 1c shows the longitudinal conductivity on $1/B$. The data exhibits well-known quantum oscillations at low fields, with the material reaching the quantum limit at ca 1.2 T. This agrees with previous measurements[15]. Interestingly, at around 7 T, thus deep in the quantum limit, an unusual additional peak in the longitudinal conductivity occurs, suggesting a field-induced enhancement of the density of states. In addition, the occurrence of the peak coincides with the onset of a decrease in the Hall resistance. After continuing to decrease, $\sigma_{xy}(B)$ eventually changes sign at $B \approx 25$ T, which signals a change from electron-type charge carriers to hole-type carriers.

As a starting point to understand all of the above features, in particular the peak appearing in the high-field conductivity, we have investigated a toy model of the low energy spectrum of $ZrTe_5$. The model is composed of all symmetry-allowed terms of the low-energy Hamiltonian up to quadratic order in momentum[15,27,28]. The parameters

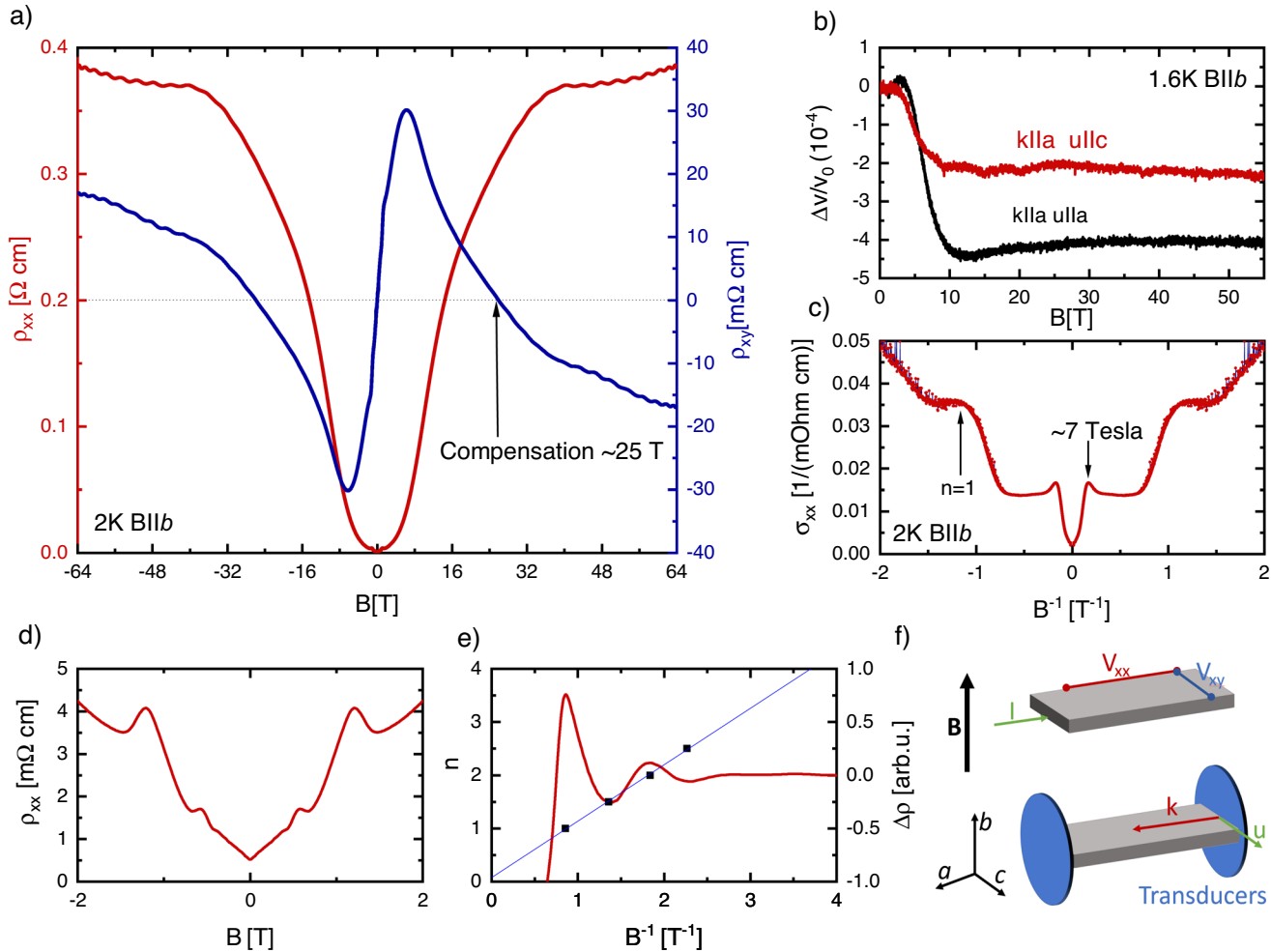

**Fig. 1 | High-field electrical charge transport in ZrTe5 at 2 K. a** Longitudinal resistivity and Hall effect measured in sample A in pulsed magnetic fields at 2 K, the arrow marks the point at which $\rho_{xy} = 0$. **b** Sound−velocity variation $\Delta v/v_0$ of a longitudinal (black) and transverse (red) sound mode propagating along the $a$-axis as a function of magnetic field applied along the $b$-axis of Sample B at 1.6 K. **c** Field dependence of the electrical conductivity of Sample A extracted by inverting the resistance tensor, displaying an anomalous maximum at ca. 7 T. **d** Shubnikov de-Haas effect measured on Sample A at 2 K in DC fields. **e** Landau-index fan diagram (black dots) combined with the magnetic field dependence of $\Delta\rho_{xx}$ of sample A. **f** Sketch of the transport-measurement configurations with respect to the three crystal axes $a$, $b$, and $c$. The electrical current $I$ and the sound waves are propagated along the $a$-axis and the magnetic field is applied along the $b$-axis.

of this model—such as the Fermi velocity, effective mass, $g$-factor and Dirac mass gap—are set by the values extracted from our experiments whenever possible, while others are expected to be small in ZrTe5. In line with earlier studies, we assume the charge density in our samples is not conserved as magnetic field strength is altered[15] and instead make the approximation that chemical potential is fixed (for a more detailed discussion, see 'Methods' and ref. 15).

Figure 2a shows the magnetic-field-induced evolution of the low-energy band structure upon including the two symmetry allowed Zeeman terms[27]. An important effect of the magnetic field is the closing of the Dirac gap, which we find happens already at a moderate field strength of about 2 T. Utilising the parameters relevant for our ZrTe5 samples[15] and the sign of the g-factor found using ab initio techniques[29], our model is consistent with a negative sign of the mass gap, which implies a strong topological insulator (TI) nature of ZrTe5[11] at low temperatures. Further increasing the magnetic field up to about 7 T enhances the energy of the hole band to a point where it crosses the chemical potential and introduces hole-like carriers. For the Hall conductivity, our theory therefore predicts a maximum around the same field strength, which is in good qualitative agreement with our experimental data. As the magnetic field is increased further, the system enters a regime with linearly crossing bands realising a 1D Weyl-

band structure. The position of the chemical potential with respect to the 1D Weyl nodes is directly dependent on the applied magnetic field. As a result, the occupation of the hole band is enhanced while the occupation of the electron band decreases. Eventually the system enters a charge-neutral state at about 24 T, at this point the majority charge-carrier type changes from electrons to holes and the Hall resistance is predicted to change sign, which again qualitatively agrees with experiment. A comparison of the calculated and experimentally determined $\sigma_{xy}(B)$ is shown in Fig. 2d. Most striking is the comparison of the calculated DOS and longitudinal conductivity from our experiment plotted as a function of $1/B$ together with the sound attenuation of the longitudinal phonon mode (Fig. 2b, c, e). At low fields below 2 T, clear signatures of quantum oscillations are visible in both the calculated DOS and measured quantities. Most importantly, the additional peak seen in $\sigma_{xx}(B)$ at high fields is well reproduced by the calculated DOS, and also the change in sound attenuation $\Delta\alpha(B)$ shows a pronounced feature close to the same field. Within our theory, we can naturally interpret this as arising when the maximum of the hole Landau band crosses the Fermi energy. Full quantitative agreement is not expected since our simple single-particle theoretical model neglects parts of the magnetic field-dependence of our model parameters. In particular, we assume that chemical potential is entirely

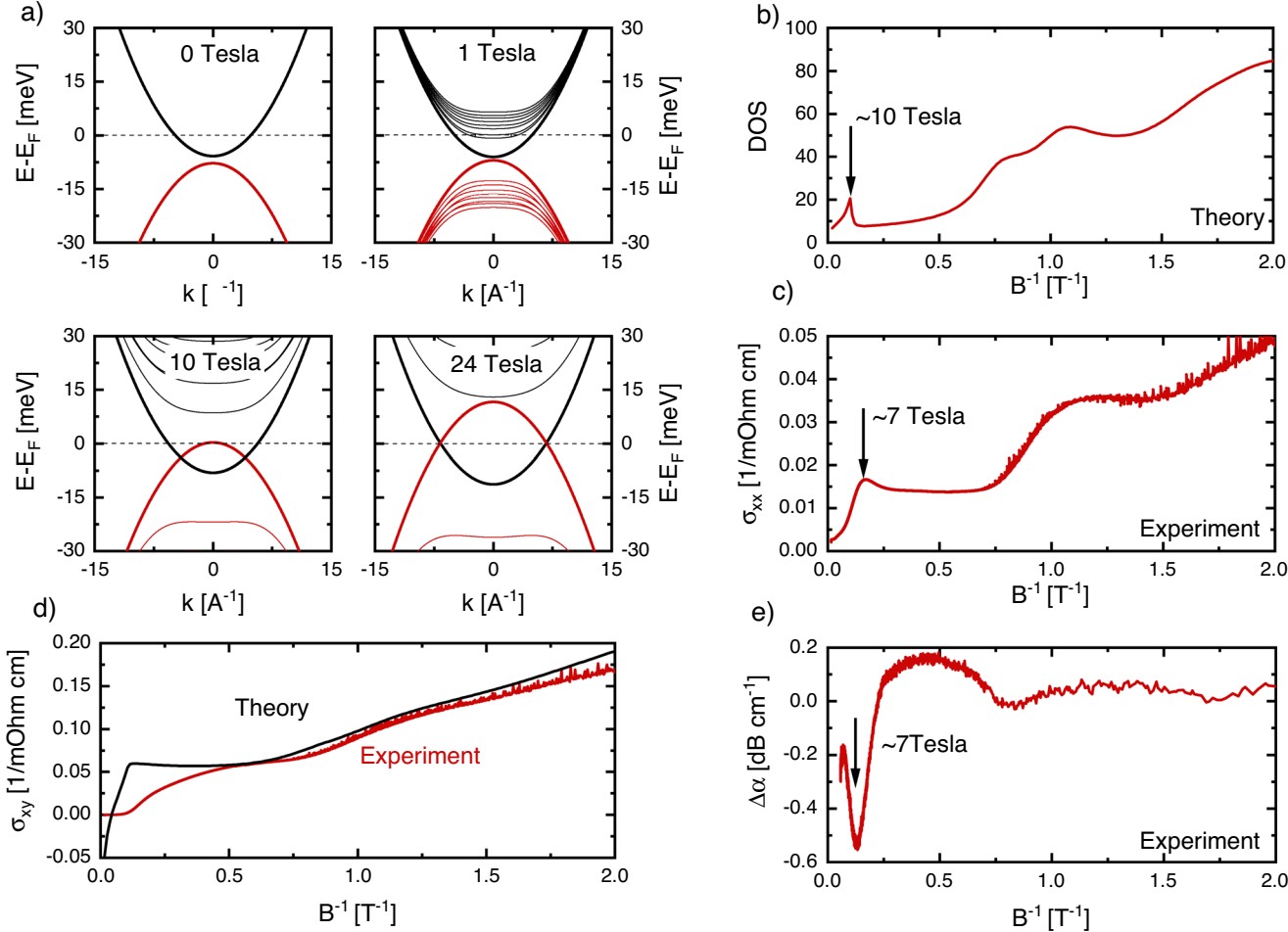

**Fig. 2 | Field-induced Lifshitz transition in ZrTe5. a** Field dependence of Landau-level dispersions with field applied along the *b*-axis. **b** Calculated density of states for the proposed model (see 'Methods' section for the parameters used). **c** Anomalous peak in electrical conductivity reflecting the expected enhancement of the DOS at high field in Sample A. **d** Comparison of measured and calculated Hall conductivity in Sample A **e S**ound attenuation of the longitudinal phonon measured reflecting the reconstruction of the Fermi surface at high field in Sample A.

fixed for all field strengths, which likely is not true at fields above ca. 2 T (in this regime, both chemical potential and particle number should exhibit a non-trivial field dependence[30]). However, the strong qualitative agreement of our simple model, explaining all UQL features found in our experiment, strongly supports its relevance to the experiment. Future experiments should nevertheless aim at independently determining the particle density as a function of magnetic field in order to improve on our modelling.

To further test our picture, we have performed magnetotransport measurements on a sample (Sample C) with the lower charge-carrier density $n = 3.8 \times 10^{16}$ cm$^{-3}$, entering the quantum limit at about 0.6 T. As a result of the lower Fermi level, one expects that the hole band should cross the chemical potential at lower fields and also that the sign change of the Hall coefficient at the charge neutrality point should occurs at lower fields than in our high-density sample. Results of our measurements on this low charge-carrier-density sample are shown in Fig. 3. Figure 3a displays the field dependence of the longitudinal magnetoresistance and Hall effect of Sample C measured at 2 K. In agreement with our expectation, we find the Hall coefficient is maximum already at a field of 5 T, and the charge neutrality point occurs at a field strength of just 8 T, a magnetic field that is easily accessible with in-house magnets. In addition, comparison of theoretical predictions of $\sigma_{xy}(B)$ and the DOS are in good qualitative agreement with experiments, similar to the samples with higher charge-carrier density.

Our findings demonstrate that ZrTe$_5$ is an excellent testbed for the largely unexplored regime of the ultra-quantum limit, which opens the door to a systematic analysis of interaction-driven instabilities at large magnetic fields and low temperatures. Our experiments, for example, reveal that a strong Zeeman splitting drives a field-induced Lifshitz transition that transforms the low-energy bands of ZrTe$_5$ into the ones of a 1D Weyl Hamiltonian. In this regime, the distance between the chemical potential and the 1D Weyl nodes is directly tuneable by the magnetic field. This opens an avenue for the investigation of interactions of 1D Weyl Fermions in the vicinity of charge neutrality with band filling being controlled via the external magnetic field. Depending on the sample, this charge neutral state occurs already around 8 T and is therefore easily accessible with commercially available magnets.

## Methods
### Sample synthesis and preparation
High-quality single-crystal ZrTe$_5$ samples were synthesised with high-purity elements (99.9% zirconium and 99.9999% tellurium), the needle-shaped crystals (about 0.1 × 0.3 × 20 mm$^3$) were obtained by the tellurium-flux method. The lattice parameters of the crystals were confirmed by single-crystal X-ray diffraction. Prior to transport measurements, Pt contacts were sputter deposited on the sample surface to ensure low contact resistance. The contact geometry was defined using Al hard masks. Prior to Pt deposition, the sample surfaces were

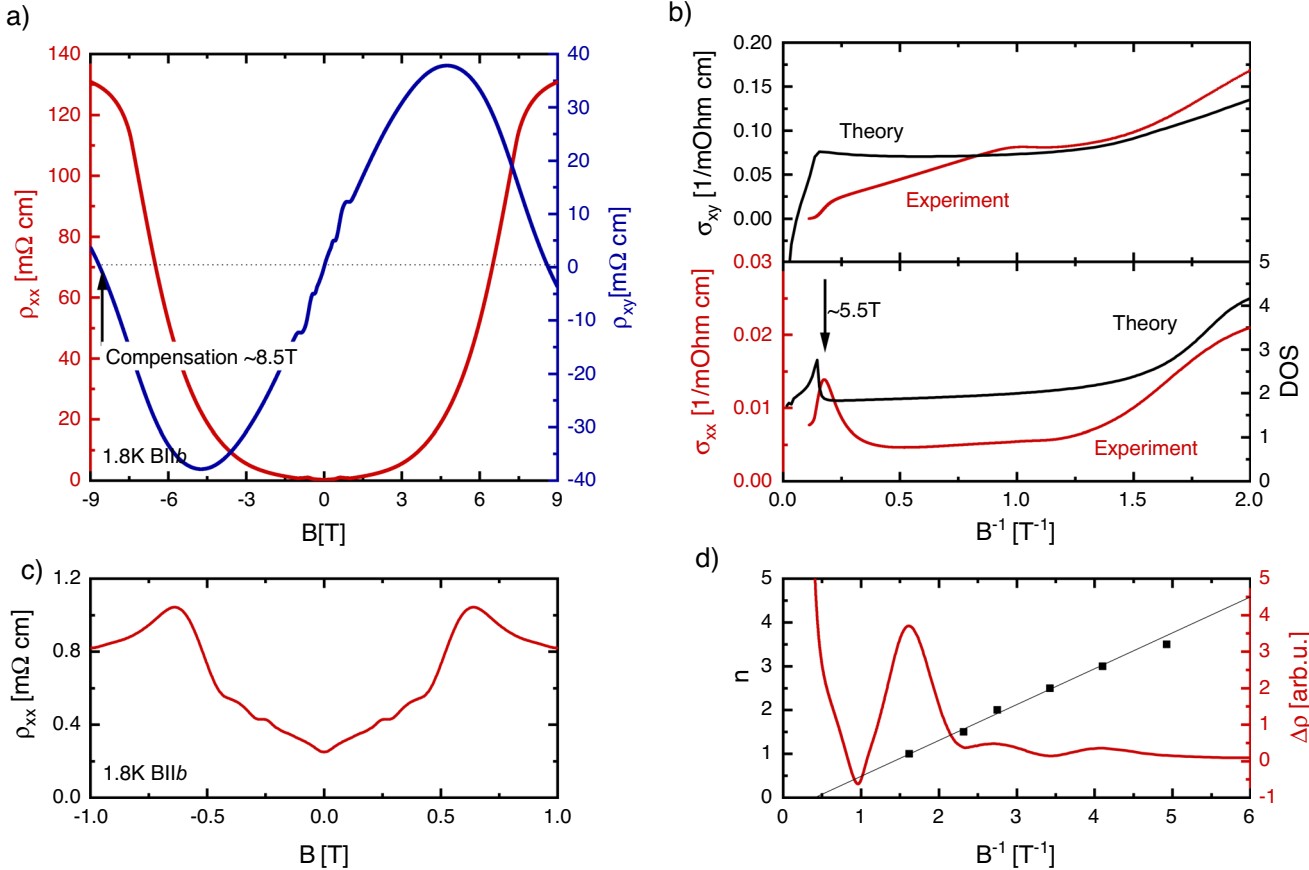

**Fig. 3 | High-field electrical-charge transport in ZrTe5 at 2 K. a** Longitudinal resistance and Hall effect measured in Sample C in static magnetic fields at 2 K, the arrow marks the point at which $\rho_{xy} = 0$. **b** Comparison of the calculated and measured Hall conductivity of sample C (upper panel) and comparison of the measured longitudinal resistance (red) and calculated DOS (black), (lower panel). **c** Shubnikov de-Haas effect measured for Sample C at 2 K in DC fields. **d** Landau-index fan diagram (black dots) combined with the magnetic-field dependence of $\Delta\rho_{xx}$ of sample C.

Argon etched and a 20-nm Ti buffer layer was deposited to ensure good adhesion of the contacts. Deposition was conducted using a BESTEC UHV sputtering system. This procedure allowed us to achieve contact resistance of the order of 1–2 Ohm.

## Sample environment
The pulsed magnetic field experiments up to 64 T were carried out at the Dresden High Magnetic Field Laboratory (HLD) at HZDR, a member of the European Magnetic Field Laboratory (EMFL). All transport measurements up to ±9 T were performed in a temperature-variable cryostat (PPMS Dynacool, Quantum Design).

## Electrical-transport measurements
The longitudinal $\rho_{xx}$ and Hall resistivity $\rho_{xy}$ were measured in a Hall-bar geometry with standard lock-in technique (Zurich Instruments MFLI and Stanford Research SR 830), with a frequency selected to ensure a phase shift below 1 degree—typically in the range between 10 and 1000 Hz across a 100 kΩ shunt resistor. In addition, some measurements were measured using a Keithley Delta-mode resistance measurement set-up for comparison. In both measurement modes, the electrical current was always applied along the $a$-axis of the crystal and never exceeded 100 μA in order to avoid self-heating.

## Ultrasound propagation measurements
Ultrasound measurements in pulsed magnetic fields up to 56 T were performed using a phase-sensitive pulse-echo technique. Two piezo-electric lithium niobate (LiNbO$_3$) resonance transducers were glued to opposite parallel surfaces of the sample to excite and detect acoustic waves. The sample surfaces were polished using a focused ion beam in order to ensure that the transducers were attached to smooth and parallel surfaces. The longitudinal and transverse acoustic waves propagated along the $a$-axis with the transverse polarisation vector along the $c$-axis. Relative sound-velocity changes $\Delta v/v$, and the sound attenuation $\Delta\alpha$, were measured for field applied along the $b$-axis. The longitudinal and transverse ultrasound propagation were measured at 28 and 313 MHz, respectively.

## Theoretical model for Hall effect and Landau quantisation in ZrTe5
To model our ZrTe$_5$ samples we use a low-energy Hamiltonian that includes all symmetry-allowed terms of the *Cmcm* group up to quadratic order in momentum[15,28]. We do not consider the small additional symmetry-breaking terms, which can result in a non-centrosymmetric phase of ZrTe$_5$ recently discovered in samples with low temperature resistivity maximum[28]. As such, near the **Γ**-point, our model reads:

$$H = \left(m + \sum_{i=a,b,c} A_i k_i^2\right)\tau^z + \hbar(v_a k_a \tau^x \sigma^z + v_c k_c \tau^y + v_b k_b \tau^x \sigma^x) + \sum_{i=a,b,c} B_i k_i^2 - \mu, \quad (1)$$

where $a, b, c$ refers to crystalline directions, $v_i$ are the Dirac velocities in these directions, $m$ is the Dirac mass, $A_i$ and $B_i$ are the prefactors for allowed terms quadratic in momentum, and the Pauli-matrices $\tau^i$ and $\sigma^i$ represent the orbital and spin degrees of freedom, respectively. We neglect the $B_i$ terms, which are expected to be small for ZrTe$_5$ since it is a Dirac semimetal. When a magnetic field, $B$, is applied parallel to the

$b$-axis the dispersion of the lowest Landau level is given by[26]

$$\varepsilon(k_b) = \pm \sqrt{\left(\frac{1}{2}\mu_B gB + A_b \mathbf{k}_b{}^2 + m\right)^2 + v_b^2 k_b^2} - \mu - \frac{1}{2}\mu_B \bar{g}B, \quad (2)$$

where $g$ and $\bar{g}$ are the $g$-factors corresponding to the two symmetry-allowed Zeeman contributions $\mathbf{H}_z = -\frac{1}{2}\mu_B gB\sigma_z$ and $\mathbf{H}_{\bar{z}} = -\frac{1}{2}\mu_B \bar{g}B\sigma_z\tau_z$, respectively. Previous experiments[15,27] on $ZrTe_5$ have shown that $v_b$ is very small and so we neglect this term; including such a term would generate a small mass gap in the 1D Weyl dispersion. Further, we assume that $A_b < 0$, and $\mu > 0$, the evolution of the dispersion described by Eq. (2) is shown in Fig. 2a. We further take the mass gap $m < 0$, which implies $ZrTe_5$ is in a strong TI phase. the corresponding density of states $\rho$ and Hall conductivity as a function of magnetic field are shown in Fig. 2b. In fact, we can describe all experimentally observed transport features in the ultra-quantum limit by the scenario depicted in Fig. 2. In particular, for fixed chemical potential, at a magnetic field strength $B_1 = \frac{2|m|}{g\mu_B}$ the gap at $k = 0$ closes and for all fields $B > B_1$ the dispersion consists of two 1D Weyl points at $k_b = \sqrt{(m + \frac{1}{2}\mu_B g|B|)/|A_b|}$. As the magnetic field is further increased the hole band intersects the Fermi level for $B_2 = 2\frac{|-m+\mu|}{(g-\bar{g})\mu_B}$ resulting in a sharp increase in the density of states—associated with an increase in the longitudinal conductivity—as well as the start of a decrease in the charge-carrier density $n$. The decreasing charge-carrier density due to the occupation of the hole band eventually results in a sign change of $n$ and, correspondingly, zero Hall conductivity at the field $B_3 = \frac{2\mu}{|g|\mu_B}$, which is the same field at which the chemical potential intersects the Weyl points of the 1D Weyl dispersion. Using parameters consistent with those found in found in ref. 15, $g = 18$, $m = -1$ meV, $A_b = -25.6$ meV nm$^2$ as well as the new fit parameter $g = -10$ and $\mu = 6.8$ meV for the sample with high charge-carrier density and $\mu = 4.5$ meV for the sample with low $n$, which are both consistent with the corresponding sample density as measured via the Hall effect, we can calculate the following fields: For the large-$n$ sample, we estimate $B_{QL} \approx 1.4$ T, $B_1 \approx 2$ T, $B_2 \approx 10$ T, and $B_3 \approx 24$ T, whereas for the low-$n$ sample $B_{QL} \approx 0.6$ T, $B_1 \approx 2$ T, $B_2 \approx 6.5$ T, and $B_3 \approx 14$ T. We also include a 0.5 meV Lorentzian broadening of the chemical potential. These values are all broadly consistent with the features observed experimentally.

## Data availability

The authors declare that all relevant data supporting the findings of this study are available within the article and its Supplementary Information. Additional data are available upon reasonable request. Source data are provided with this paper.

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

## Acknowledgements

We would like to thank T. Helm for his support in performing the pulsed field magnetotransport measurements. T.M. acknowledges funding by the Deutsche Forschungsgemeinschaft via the Emmy Noether Programme ME4844/1-1. C.F. acknowledges the research grant DFG-RSF (NI616 22/1): Contribution of topological states to the thermoelectric properties of Weyl semimetals and SFB 1143. P.M.L., G.D.G., and Q.L.

were supported by the US Department of Energy, Office of Basic Energy Science, Materials Sciences and Engineering Division, under contract DE-SC0012704. We acknowledge support from the DFG through the Würzburg-Dresden Cluster of Excellence on Complexity and Topology in Quantum Matter – *ct.qmat* (EXC 2147, project-id 39085490), the Collaborative Research Center SFB 1143 (Project No. 247310070), the ANR-DFG grant Fermi-NESt, and by Hochfeld-Magnetlabor Dresden (HLD) at HZDR, member of the European Magnetic Field Laboratory (EMFL). J.G. acknowledges support from the European Union's Horizon 2020 research and innovation programme under Grant Agreement ID 829044 "SCHINES". We acknowledge DESY (Hamburg, Germany), a member of the Helmholtz Association HGF, for the provision of experimental facilities. H.F.L. acknowledges the support of the Georg H. Endress foundation. The work at BNL was supported by the US Department of Energy, office of Basic Energy Sciences, contract no. DOE-sc0012704

## Author contributions

S.G. and J.G. conceived the experiment. The single crystals were grown by G.D.G. and P.M.L. S.G., R.W., and C.F. fabricated the final transport devices. S.G., R.W., and T.F. performed the transport experiments. S.G. and M.U. performed Seebeck measurements, S.G., M.K., D.G., S.Z., and J.W. performed ultrasound-propagation measurements. S.G. and T.F., performed high-field transport measurements. H.F.L. and T.M. provided the theoretical model. S.G., H.F.L., and J.G. analysed the data. All authors contributed to the interpretation of the data and to the writing of the manuscript.

## Funding

## Competing interests

The authors declare no competing interests.
