## [Peer Review File · Nature Communications]

Signatures of a magnetic-field-induced Lifshitz transition in the ultra-quantum limit of the topological semimetal ZrTe_5REVIEWER COMMENTS

Reviewer #1 (Remarks to the Author):

It is well known that the quantum limit cannot be easily reached in most materials. Recent years have witnessed the research focus in ZrTe5 and HfTe5 sister materials, and the lower temperatures of achieving the quantum limit of these two materials. Different from other materials, the materials of ZrTe5 are extremely sensitive to the growing environments, e.g. forming strong topological insulators or weak topological insulators. In particular, the three-dimensional quantum Hall effect was first reported in the ZrTe5 with the weak topological insulator phase at the extreme quantum limit in the presence of weak magnetic field about several tesla. This three-dimensional quantum Hall effect together with other studies focused on ZrTe5 made ZrTe5 an ideal platform to investigate various interaction-driven topological phases.

In this work, the authors have experimentally characterized the quantum limit of ZrTe5 at fields up to 64 T by a combination of electrical-transport and ultrasound measurements, and found that the Zeeman effect in ZrTe5 enables an efficient tuning of the 1D Landau band structure with magnetic field.

The experimental findings in this work are clearly convincing and the corresponding explanations are reasonable. And a theoretical analysis is also included to further understand the experimental findings, which is good for readers to understand.

In addition, the work is timely, though there have been lots of studies on the topics related to ZrTe5. In summary, the referee would believe that the present manuscript warrants to be accepted in Nature Communications after the authors can properly address the following comments.

The minor comments are that in the whole manuscript the author did not review the part of three-dimensional quantum Hall effect that is the striking transport property of ZrTe5, and the statement of "establishment of ideal platform for interaction-driven topological phases" is over self-praising, which has actually demonstrated in Ref. [19].

Reviewer #2 (Remarks to the Author):

In this manuscript, S. Galeski and co-authors carried out systematically studies on ZrTe5 in the ultra-quantum limit. They found the field-induced Lifshitz transition and charge-neutral state in this regime, using magneto-transport and ultrasound measurements. The calculated DOS is consistent with the experimental data in their theoretical model, including the longitudinal conductivity and sound attenuation. All of the measurement data are solid, however, the explanation of the results is not convincing. Below are the questions.

1. For the Hall conductivity, the theory predicts a maximum around 7T, but the experiment data isn't consistent with the theory around the same magnetic field. What is the reason?
2. Although the theory is in good qualitative agreement with the longitudinal conductivity and sound attenuation, the transport evidence of the transition and charge-neutral state is weak and not enough. More experiments, like angle-dependent magneto-transport and thermoelectric measurement, should be carried out to prove that.
3. In Figure S2, the linear fit of sample C seems not proper because of the narrow fitting range, which may result in a wrong calculated carrier density. The author should give a reason for the chosen range. In addition, the R-T curve of sample A is needed to see more differences between samples A and C.
4. The influence of temperature should be considered. The author should show the temperature dependence of the transition field and analyze it in their theoretical model.

Reviewer #3 (Remarks to the Author):

The manuscript by S. Galeski et al. reports the electrical transport and elastic properties of ZrTe5 under high magnetic fields up to 65 T and interpreted the results by theoretical calculation based on the Dirac model. The authors observed significant anomalies at ~10 T in the conductivity, sound velocity, and attenuation coefficient. Besides, no indication of the phase transition was observed in the thermodynamic quantities above 10 T, ruling out the formation of density wave states in the high-field region. Based on a simple model calculation, the authors proposed that the lowest hole Landau level crosses the Fermi level at 10 T, which is driven by the Zeeman effect. This scenario is supported by the accordance of theoretical DOS and Hall conductivity with their experimental results.

Their findings showed that ZrTe5 can be a platform to investigate the ultra-quantum-limit state under a moderate magnetic field, whose electronic structure can be continuously controlled by an external magnetic field via the Zeeman effect. The authors' new interpretation is well supported, and such a clean 1D Weyl system with flexible tunability via magnetic field is of physical interest. The referee can recognize novel insights into the topological electronic states realized in ZrTe5. However, the authors should properly address or answer the following concerns/questions before the final decision.

(1) The authors deduced the carrier density (n) from the initial slope of the Hall resistivity based on the single-carrier model, but I am concerned about whether this is reasonable. According to the authors' previous study, the shape of the Fermi surface at 0 T is regarded to be an ellipsoid. Can the authors estimate the carrier density from the Fermi surface volume? Is it consistent with n estimated from Hall measurement?

(2) In the theoretical model mentioned after line 241, the author is unclear about which quantities are already known from the previous experiments, and which are the adjustment parameters. The authors should thoroughly describe the used parameters with necessary references.

(3) In the figures, it seems that the Hall resistivity ρ_{xy} and Hall conductivity σ_{xy} are defined as positive (negative) when the electron (hole) carrier is dominant. However, the referee feels it is the opposite definition compared to the ordinal one. To avoid confusion, the referee recommends changing the definition of these quantities. If the authors are willing to use the present notation, a comment should be added to the main text.

(4) In Fig. 2d and 3b, experimental σ_{xy} seems to be reasonably reproduced by calculation in the low field region below 1 T, however, the mismatch becomes significant in the high-field region. What is the origin of this discrepancy? Regarding this point, Does the present calculation of σ_{xy} consider the contribution from the Berry curvature? (I think it is better to describe the details of the σ_{xy} calculation) I assume that the lowest Landau levels form a 1D Weyl band structure, and thus, I suppose there exists a nontrivial anomalous Hall contribution. Can the author comment on whether this effect is important or negligible in the present case?

(5) In Fig. 2e, the relative attenuation coefficient is negative at the level-crossing fields (e.g. $B \sim 1$ and 7 T). This means that the ultrasound is less absorbed at the level-cross, is this correct? On the other hand, In the well-known acoustic dHvA effect such as elemental semimetal Bi [please see J. Phys. Soc. Jpn. 21, 1744 (1966), Sci. Rep. 9, 1672 (2019), etc.], the attenuation is enhanced when each Landau tube passes through the Fermi surface [please see a book D. Shoenberg, "Magnetic oscillations in metal"]. Why is the present behavior opposite compared to the ordinal acoustic dHvA?

Dear Referee 1,

thank you for your review. We find your appreciation of our work and the recommendation to publish in Nature Communications very gratifying. In the following, we would like to address the points that were raised in the review in the same sequence:

- 1) *The minor comments are that in the whole manuscript the author did not review the part of three-dimensional quantum Hall effect that is the striking transport property of ZrTe5 [...] has actually demonstrated in Ref. [19].*

We thank the reviewer for bringing our attention the lack of a detailed discussion of the 3D quantum Hall effect. Since the present manuscript focuses on the physics deep in the quantum limit, we have largely retained a focus on this regime and the relation of our findings to previous transport literature on the subject. However, we agree that the appearance of plateaus in Hall resistance of ZrTe5 is perhaps one of the most striking discoveries in the field. In the current version of the manuscript, we have modified the introduction to more prominently discuss the effect and credit the work cited by the referee.

- 2) *the statement of “establishment of ideal platform for interaction-driven topological phases” is over self-praising [...].*

We have modified this sentence in the current version of the manuscript.

Sincerely yours,

Authors

Dear Referee 2,

thank you for the elaborate and detailed review. In addition, we are happy for your appreciation of our experimental work. In the following, we would like to address the points that were raised in the review. We hope that the new measurements and the modifications we have made in response to the reviewer's comments clarify that our explanation of the observed transport behavior in the ultra-quantum limit of ZrTe5 is accurate and convincing.

1) For the Hall conductivity, the theory predicts a maximum around 7T, but the experiment data isn't consistent with the theory around the same magnetic field. What is the reason?

We are grateful to the reviewer for allowing us to clarify this point, which was not fully discussed in the original version of the paper. We would first like to emphasize that the theoretical model is intended to provide a qualitative description of the transport measurements in ZrTe5, and not reproduce precisely the quantitative values of Hall conductivity. In particular, the key features of the transport data, namely the sign change of the Hall conductivity and the additional peak in the longitudinal conductivity, are explained by our simple model.

That said, the reviewer is completely correct that the extended plateau in σ_{xy} up to $B \sim 7$ T in the theory is not observed in experiment. The primary reason for this is that the theory assumes that the chemical potential is completely fixed as magnetic field is increased, if instead density were fixed, one expects the simple dependence $\sigma_{xy} \sim 1/B$ at large fields.

Although a plateau is observed in the experimental data - indicating some deviation of the particle number from its low-field - this plateau only extends to $B \sim 2$ T. This suggests that, in reality, both the chemical potential and density change as a function of magnetic field and neither is completely constant. Allowing for this dependence of the chemical potential on magnetic field could produce an improved fit, but would introduce an additional ad-hoc fitting parameter. Since we currently have no means of independently determining the chemical potential at arbitrary magnetic fields, we decided against a model with an additional, uncontrolled fitting parameter - this in our view is the more honest calculation to do. In response to the referee's question, we have added a comment on the limit of constant chemical potential in the main text.

The fact that neither density nor chemical potential are completely fixed is further confirmed by the fact that the theoretical Hall conductivity at large magnetic fields is dominated by the hole band taking large negative values, in contrast the experimental Hall conductivity - whilst exhibiting the predicted change of sign - has a magnitude that remains relatively small, which is the expected behavior when neither chemical potential nor density are completely fixed but there is a change of carrier type.

2) Although the theory is in good qualitative agreement with the longitudinal conductivity and sound attenuation, the transport evidence of the transition and charge-neutral state is weak and not enough. More experiments, like angle-dependent magneto-transport and thermoelectric measurement, should be carried out to prove that.

Although we believe the presented set of measurements were sufficient proof of our claims, we fully agree that delivering further evidence could strengthen the message of our work. Thus, we have performed the requested additional measurements. Since the field scales necessary to access the relevant physics in the high carrier density samples exceed the magnetic fields available for common

DC magnets (measurements of the Seebeck effect in pulsed fields do not seem plausible) in the following we will have focused our attention on samples with a small change carrier density.

We agree with the referee that measurements of the Seebeck effect could be an excellent to confirm the change of carrier type and thus emergence of the field induced charge neutral state. Unfortunately, during our attempts to prepare sample C for thermoelectric measurements it broke. Due to this we have performed the measurements on another sample from the same batch now referred to as sample D. Magnetotransport (included in the SI) measurements on sample D revealed it harbored a similar charge carrier density to sample C and exhibited qualitatively similar behavior in charge transport. Result of our Seebeck measurements is presented below (now also included in the SI).

Figure 1 Field dependence of the Seebeck effect measured in sample D at 550mK

As could be expected from our analysis, the Seebeck signal changes sign similar to the Hall effect at ca. 10T. Here the field seems marginally higher than in the case of the Hall effect, however we attribute this to a small misalignment of the sample b-axis with the magnetic field. The reason is that, in order to build up a measurable gradient to measure the electric response we have installed the sample in a semi-suspended configuration (see inset) and the weight of thermometers and heaters have slightly bent it downward.

This explanation becomes plausible on investigating the angular dependence of the Hall effect. As can be seen in figure 2a a small misalignment of ~ 5 degrees in the a-b plane already moves the compensation point beyond the measurable range of our standard DC magnet (9 T).

Figure 2: Angular dependence of Hall effect in sample D. Magnetoresistance versus magnetic field B for various rotation angles in the a. a-b plane b. c-b plane.

This observation is in good agreement with the known anisotropy of the effective g -factors factors found in ZrTe₅ [ref 28]. In particular it is believed that both Zeeman terms are strongest when the b -axis and the magnetic field are coaligned. Rotating the field away from the b -axis thus is expected to push the charge compensation point away to higher fields as seen in our angular study of the Hall effect. In addition to the results presented here we have now in addition extended the SI with angle dependent magnetoresistance data measured on samples A and D.

3) In Figure S2, the linear fit of sample C seems not proper because of the narrow fitting range, which may result in a wrong calculated carrier density. The author should give a reason for the chosen range. In addition, the R-T curve of sample A is needed to see more differences between samples A and C.

The principal reasons behind choosing such a narrow fitting range are the appearance of first quantum oscillations (QOs) in the Hall effect already at those small fields, thus in order to avoid the influence of QO on the resulting fitting we have confined ourselves to a narrow fitting range where the Hall resistance is smooth without any signatures of QO. This discussion is now included in the appropriate section of the SI.

In addition, we have now estimated the carrier density for both small and large FS samples (A and D) using the calculated FS volume assuming an elliptical FS. Those calculations are in reasonable agreement with values extracted from the Hall effect confirming a smaller carrier density of sample D (we were not able to perform the angular magneto transport study of sample C since it broke during our first attempts to measure the Seebeck effect, however being from the same batch and exhibiting similar qualitative behavior and very close value of n_{Hall} we believe they can be treated as equivalent.)

We have now included the R(T) of sample A and sample D in Fig. S1 of the SI.

4) The influence of temperature should be considered. The author should show the temperature dependence of the transition field and analyze it in their theoretical model.

Experimentally the temperature dependence of the movement of the compensation point could be seen clearly in the temperature dependence of Hall resistance measured in samples A and C Figures S4b and S5B with the compensation point moving to higher fields at higher temperatures. However,

modelling of the temperature dependence of phenomena is a non-trivial matter. The difficulty arises from dependence of the chemical potential of ZrTe_5 on temperature.

It has been shown that in ZrTe_5 the chemical potential drastically shifts with changing temperature leading to the most spectacular charge transport feature observed in ZrTe_5 – the huge resistance peak attributed to the Lifshitz transition where the chemical potential crosses from the electron to the hole band. Although there is a growing body of works discussing the temperature dependence of the chemical potential, its physical origin is not yet clear.

In these circumstances it seems that although we could ‘fit’ our model to the data at elevated temperatures it would not provide any additional information. In this case we believe restraining our model to the effective ‘zero temperature’ region where the charge carrier density does not considerably change is the most clear and informative choice.

Again, we thank you for your suggestions how to improve our manuscript. We hope that our the additional measurements and further explanations clarified all your concerns and that you find the modified version of the manuscript to be up to the standard that would merit publication in Nature Communications

Sincerely yours,

Authors

Dear Referee 3,

thank you for the elaborate and detailed review. We are very satisfied with your recognition of the novelty and quality of our work. In the following we will address the points that were raised in the review.

- 1) *The authors deduced the carrier density (n) from the initial slope of the Hall resistivity based on the single-carrier model, but I am concerned about whether this is reasonable. According to the authors' previous study, the shape of the Fermi surface at 0 T is regarded to be an ellipsoid. Can the authors estimate the carrier density from the Fermi surface volume? Is it consistent with n estimated from Hall measurement?*

The principal reasons behind choosing a single carrier model for our fits originates from the fact that according to ARPES and DFT studies of ZrTe₅ for our carrier density all additional bands: the hole band and the nearest quadratic electron band located in corner in the BZ are ca ~ 100 K away from the Fermi level at low fields and temperatures and thus are not expected to contribute to transport at low temperatures < 2 K. However, for the purpose of this argument we have performed additional measurements of the SdH effect along principal axes of samples A and D (we were not able to perform additional measurements on sample sample C since it broke during our attempt to measure the Seebeck effect, however being from the same batch and exhibiting similar qualitative behavior and a very close value of n_{Hall} we believe they can be treated as equivalent).

From those measurements we have estimated the carrier density for both small and large FS samples (A and D) using the calculated FS volume assuming an elliptical FS. The resulting values: $n_{\text{Hall, A}} = 6.1 \cdot 10^{16}$, $n_{\text{Volume, A}} = 4.1 \cdot 10^{16}$ and $n_{\text{Hall, D}} = 3.9 \cdot 10^{16}$, $n_{\text{Volume, A}} = 2.2 \cdot 10^{16}$, seem to be in a reasonable agreement justifying the use of the 1 carrier model at small magnetic fields.

- 2) *In the theoretical model mentioned after line 241, the author is unclear about which quantities are already known from the previous experiments, and which are the adjustment parameters. The authors should thoroughly describe the used parameters with necessary references.*

We thank the reviewer for highlighting this. The majority of these parameters come from our previous study of ZrTe₅ (S. Galeski, Nat Commun 12, 3197 (2021)). We have modified our discussion of the parameters highlighting which parameters are known from this previous study and which parameters are inferred from the current experiment.

- 3) *In the figures, it seems that the Hall resistivity ρ_{xy} and Hall conductivity σ_{xy} are defined as positive (negative) when the electron (hole) carrier is dominant. However, the referee feels it is the opposite definition compared to the ordinal one. To avoid confusion, the referee recommends changing the definition of these quantities. If the authors are willing to use the present notation, a comment should be added to the main text.*

Thank you for pointing out the issue. We agree that commonly used convention is different than the one used in our manuscript. We have now added an additional comment on this to the manuscript. In addition, we have noticed that there was an inconsistency in definitions of the sign of the Hall effect between figure 1 and 2. This has also been fixed.

- 4) *In Fig. 2d and 3b, experimental σ_{xy} seems to be reasonably reproduced by calculation in the low field region below 1 T, however, the mismatch becomes significant in the high-field region. What is the origin of this discrepancy? Regarding this point, Does the present calculation of σ_{xy} consider the contribution from the Berry curvature? (I think it is better to describe the details of the σ_{xy} calculation) I assume that the lowest Landau levels form a 1D Weyl band structure, and thus, I suppose there exists a nontrivial anomalous Hall contribution. Can the author comment on whether this effect is important or negligible in the present case?*

The model that we have used in our work was selected based on its simplicity. It is the minimal model that correctly describes the qualitative features of our data, namely the sign change of the Hall conductivity and the additional peak in the longitudinal conductivity. Although our model does fully include the Berry curvature of the Landau bands in the calculation σ_{xy} there are major simplifications that can contribute to the discrepancy between the measured and calculated σ_{xy} .

We believe that the main reason for the discrepancy originates from the simplified treatment of the chemical potential since in our calculation we assume that the chemical potential is completely fixed. This assumption is validated by the fact that if instead density was fixed, one would expect a simple dependence $\sigma_{xy} \sim 1/B$.

This assumption works very well in the low field regime up to 2Tesla as we have previously shown (ref 19). However deep in the quantum limit likely in reality, both the chemical potential and density change as a function of magnetic field and neither is completely constant. Allowing for this dependence of the chemical potential on magnetic field could produce an improved fit, but would introduce an additional ad-hoc fitting parameter. Since we currently have no means of independently determining the chemical potential at arbitrary magnetic fields, we decided against a model with an additional, uncontrolled fitting parameter - this in our view is the more honest calculation to do. In response to the referee's question, we have added a comment on the limit of constant chemical potential in the main text.

The fact that neither density nor chemical potential are completely fixed is further confirmed by the fact that the theoretical Hall conductivity at large magnetic fields is dominated by the hole band taking large negative values, in contrast the experimental Hall conductivity - whilst exhibiting the predicted change of sign - has a magnitude that remains relatively small, which is the expected behavior when neither chemical potential nor density are completely fixed but there is a change of carrier type.

An additional reason for the discrepancy in the vicinity of the charge neutrality point could originate from the unscreened Coulomb interaction. Our calculation is essentially a single particle model. However, in ZrTe5 already at 0 field the screening length exceeds 2-unit cells for the discussed electronic densities. This effect will become even stronger close to the neutrality point where the screening length is expected to diverge. Additionally, the dimensional reduction in strong magnetic fields leads to quenching of particle motion in the a-c plane further enhancing the effects of interactions.

In fact, we believe that the last point makes our current work particularly interesting since to our knowledge there are very few, if any systems other than ZrTe5 where such physics can be studied.

- 5) *In Fig. 2e, the relative attenuation coefficient is negative at the level-crossing fields (e.g. $B \sim 1$ and 7 T). This means that the ultrasound is less absorbed at the level-cross, is this correct? On the other hand, In the well-known acoustic dHvA effect such as elemental semimetal Bi [please see J. Phys. Soc. Jpn. 21, 1744 (1966), Sci. Rep. 9, 1672 (2019), etc.], the attenuation is enhanced when each Landau tube passes through the Fermi surface [please see a book D. Shoenberg, "Magnetic oscillations in metal"]. Why is the present behavior opposite compared to the ordinal acoustic dHvA?*

We thank the referee for their excellent question. The textbook cited by the referee summarizes concisely that ultrasound propagation along the direction of the Landau tubes peaks shortly before a Landau tube crosses the Fermi energy. More precisely, the peak arises when electrons in the respective Landau tube propagate with the same speed as the phonons. Our experiments, however, are not in this setup. Rather, they are performed with the magnetic field perpendicular to the direction of ultrasound propagation. This means that the "surf riding"-picture does not apply.

It is worth noting that measured ultrasound attenuation quite generally does not track the theoretical DoS, nor the experimental conductivity (both have similar features highlighting the fields at which Landau tubes cross the Fermi energy). This in our view confirms that the picture proposed by Shoenberg is not directly applicable for our setups.

Alternatively, one might think about ultrasound attenuation along the lines of PRB 104, 245117 (2021) by some of the present authors. In qualitative agreement with Shoenberg's picture, this model also predicts strong features of the ultrasound attenuation at the fields at which the Landau band bottoms cross the Fermi energy.

While we do at the moment not have a microscopic quantitative explanation of the ultrasound signal, we do think that the dip in ultrasound attenuation at ca. 7T is related to the nearby crossings of the Landau band bottom and the Fermi level. The dip ultrasound attenuation would for example be consistent with the magnetic field moving away from values at which a resonance condition related to the Landau tube crossing the Fermi energy is fulfilled.

Given the above, we agree with the referee that our previous wording was inadequate, and thank them for pointing this out. We have now carefully rewritten the manuscript to state that the peak in α is related to, but not in its position directly marking, the crossing of the hole band and the Fermi energy.

Again, we thank you for your suggestions how to improve our manuscript. We hope that our further explanations clarified all your concerns and that you find the modified version of the manuscript to be up to the standard that would merit publication in Nature Communications

Sincerely yours,

Authors

REVIEWER COMMENTS

Reviewer #2 (Remarks to the Author):

I appreciate the additional experiments and answers that the author responded to these questions. However, there are still some problems.

1. The author claims that the limit of constant chemical potential leads to the disagreement between theory and experiment data of Hall conductivity, especially in the high field region. Does this limit influence the calculated DOS at the high field? What's the reason that the longitudinal conductivity can fit the model well while the Hall conductivity can not?
2. The data of thermoelectric measurement is noisy, and in the small field region, there is some feature that also shows sign change, which makes the explanation not convincing. I think the author should make more discussion on the thermoelectric experiment data.
3. In fact, a similar magnetic-field-induced phenomenon [Nature Physics 13, 979-986(2017)] has been reported. The author should show their significance and contribution to this research field more clearly to match the high level of this journal.

Reviewer #3 (Remarks to the Author):

The authors have satisfactorily answered all my queries/concerns, and the revisions made are reasonable. Now I think that the manuscript is suitable for publication in Nature Communications.

Dear Referee 2,

Thank you for the additional comments and detailed study of our manuscript and responses. In the following we will fully address the remaining points in the order they were raised.

- 1. The author claims that the limit of constant chemical potential leads to the disagreement between theory and experiment data of Hall conductivity, especially in the high field region. Does this limit influence the calculated DOS at the high field? What's the reason that the longitudinal conductivity can fit the model well while the Hall conductivity can not?*

As shown in Fig. R1, a completely fixed chemical potential leads to large changes in density, n , as a function of magnetic field, especially in the lowest Landau level. In contrast, a fixed density would lead to large changes in chemical potential and does not allow for the change of the sign of Hall conductivity that is observed experimentally. In practice, however, it is most likely that neither chemical potential nor density are completely fixed (this is evidenced by the appearance of the quasi-quantized Hall effect).

To model such a scenario we have used a chemical potential profile that limits the large changes in density/chemical potential, see Fig. R1 and R2. When such a profile the Hall conductivity is closer to the experimental value and the DOS peak also occurs closer to the experimental value of ~ 7 T (see R3 and R4). This shows that an intermediate scenario between fixed chemical potential and fixed density can very well explain all our data. Hence, the discrepancy between theoretical and experimental Hall conductivity for fixed chemical occurs because of the large changes in density/chemical potential this would entail.

That said, we want to stress that it was not our intention to provide a quantitative fit of the experimental findings, which requires an assumption about the chemical potential profile. Rather the aim of our theoretical model was to explain the key qualitative features in our experiment: the change of sign in Hall conductivity and the peak in longitudinal conductivity. The model with fixed chemical potential is able to explain both features with a reasonable degree of accuracy, with the minimum number of free parameters, and without any ad-hoc assumption about the chemical potential profile.

- 2. The data of thermoelectric measurement is noisy, and in the small field region, there is some feature that also shows sign change, which makes the explanation not convincing. I think the author should make more discussion on the thermoelectric experiment data.*

Thank you for pointing out this detail. We attribute the additional feature appearing at low field to a brief temperature instability appearing due to the magnetocaloric effect when the magnetic fields change polarity.

In this measurement we were primarily interested in the high field region where such artifacts do not occur and thus, we decided to conduct the experiment slowly but continuously sweeping the magnetic field (9.8 Oe/s). In order to avoid this artifact in the low field region, the magnetic field during the measurement should be stabilized for each measurement point and then one should wait for stabilization of the sample's temperature before recording data.

We would like to stress that the low field Seebeck effect of ZrTe_5 has been studied in detail and data from multiple groups including the present authors has been published: ref 15, PRL 123 196602, PRB 103 045203, npj Quantum Mater. 7, 71 (2022), Nat Comm 11, 1046 (2020) to list a few. None of that report such an anomalous low field change of sign in the Seebeck effect, additionally confirming that the feature is not of intrinsic origin.

We have now added a comment regarding this additional feature to the appropriate section of the supplement.

3. *In fact, a similar magnetic-field-induced phenomenon [Nature Physics 13, 979-986(2017)] has been reported. The author should show their significance and contribution to this research field more clearly to match the high level of this journal*

Thank you for pointing out this important omission in our literature review. We have now included it in the discussion of our findings.

We would like to again thank you for suggestions how to improve our manuscript. We hope that our additional explanations clarified all remaining concerns. We hope you will now find the modified version of the manuscript up to the standard of Nature Communications

Sincerely yours,

Authors

Figure R1

Figure R2

Figure R3

Figure R4

Fig. R1-2: *Density and chemical potential profiles as a function of magnetic field.* Fixed chemical potential (blue) results in a large change of density at large magnetic field, where the hole band dominates. Conversely fixed density (orange) results in a large change of chemical potential at large magnetic fields. An intermediate profile of chemical potential (green), here taken as the combination of 75% of the fixed density profile and 25% of the fixed chemical potential profile, reduces both effects at large field. Fig. R3-4: *Hall conductivity and DOS as a function of magnetic field.* The Intermediate profile (green) more closely tracks the experimental Hall conductivity and allows for both a change of sign in Hall conductivity at large magnetic fields, as well as the increase of DOS at intermediate magnetic fields. The increase in DOS for a fixed density (or partially fixed) is larger than for fixed chemical potential since the DOS increases when the chemical potential is close to the bottom of the electron Landau band. We note that the exact relationship between DOS and longitudinal conductivity is unclear since the latter will include other B dependent terms, for instance through the full dependence of the transport lifetime and Fermi-velocity on magnetic field.

REVIEWER COMMENTS

Reviewer #2 (Remarks to the Author):

I appreciate the answers that the author responded to my questions. However, I think the answers don't remove my concerns. For the second question, the author claims that the temperature instability leads to the sign change of the low field Seebeck signal. But the amplitude of the low and high field signal is almost of the same order. The author's explanation makes me more concerned about the reliability of their Seebeck signal. As for the third question, as I mentioned, the findings of this manuscript are similar to the reported phenomenon [Nature Physics 13, 979-986(2017)], which reduces the overall novelty and significance of the manuscript, and the author's answer to this question doesn't address the novelty issue. To sum up, I think that the manuscript doesn't reach the high level of this journal.

Dear Referee 1,

thank you for your final comments. In the following, we would like to address the points that were raised in the review in the same sequence:

- 1) For the second question, the author claims that the temperature instability leads to the sign change of the low field Seebeck signal. But the amplitude of the low and high field signal is almost of the same order. The author's explanation makes me more concerned about the reliability of their Seebeck signal.

We do not share the Referees concern. As we have pointed out the field range that the referee finds worrying has been measured previously with care both by us and other groups corroborating our interpretation that the anomalous low field feature found in our data is a measurement artifact. In particular in npj Quantum Mater. 7, 71 (2022) authors have measured the Seebeck effect in samples grown under the same conditions as ours with similar charge carrier density.

Although the discussion focuses on a minor detail of supplementary data and not on the main findings of our work, we would like to finally clarify the issue by presenting of one of our preliminary measurements done on the same sample in our in-house 9T cryostat. Here the magnet controller was allowing for sweeping the field at a much smaller rate - 0.5 Oe/s. Inspection of the data reveals that the discussed feature is completely absent, in agreement with data found in literature.

- 2) As for the third question, as I mentioned, the findings of this manuscript are similar to the reported phenomenon [Nature Physics 13, 979-986(2017)], which reduces the overall novelty and significance of the manuscript, and the author's answer to this question doesn't address the novelty issue. To sum up, I think that the manuscript doesn't reach the high level of this journal.

We kindly yet firmly disagree. Although both Nature Physics 13, 979-986(2017) and our work describe the change of the sign of Hall effect, the physics of both situations is very different. In the Nat Phys article authors discuss gapping and annihilation of one type of Weyl nodes in the Weyls semimetal TaP. In stark contrast, our work focuses on ZrTe5 that harbors a single massive Dirac electron pocket. In our case there are no Weyl crossings present at all. Thus, the mechanism of coupling of Weyl fermions of opposite chirality due to strong magnetic fields is not at all relevant.

In summary although both Us and Authors of the Nat Phys article utilize similar experimental techniques the studied material systems harbor dramatically different band structures and the appearance of the change of the Hall sign is a manifestation of very different physical phenomena.

Sincerely yours,

Authors